# Learning to Trust: Bayesian Adaptation to Varying Suggester Reliability in Sequential Decision Making

## Abstract

Autonomous agents operating in sequential decision-making tasks under uncertainty can benefit from external action suggestions, which provide valuable guidance but inherently vary in reliability. Existing methods for incorporating such advice typically assume static and known suggester quality parameters, limiting practical deployment. We introduce a framework that dynamically learns and adapts to varying suggester reliability in partially observable environments. First, we integrate suggester quality directly into the agent's belief representation, enabling agents to infer and adjust their reliance on suggestions through Bayesian inference over suggester types. Second, we introduce an explicit "ask" action allowing agents to strategically request suggestions at critical moments, balancing informational gains against acquisition costs. Experimental evaluation demonstrates robust performance across varying suggester qualities, adaptation to changing reliability, and strategic management of suggestion requests. This work provides a foundation for adaptive human-agent collaboration by addressing suggestion uncertainty in uncertain environments.

## 1 Introduction

Autonomous agents operating in uncertain environments can greatly benefit from external guidance provided by humans or automated systems. For example, search and rescue robots might rely on human operators who suggest promising search locations, while autonomous vehicles can use passenger alerts about hazards not detected by onboard sensors. However, real-world suggestions vary in reliability due to factors such as human fatigue, sensor degradation, or changing environmental conditions. Thus, agents must dynamically learn how much trust to place in external suggestions and strategically decide when to seek guidance.

Shared autonomy research has extensively explored human-agent collaboration, often focusing on intent inference and control blending between agents and humans (Dragan & Srinivasa, 2013; Javdani et al., 2018). Typically, these systems assume static or known reliability models, limiting their adaptability to changing conditions (Losey & O'Malley, 2019). Recently, action suggestions have been treated as environmental observations within partially observable Markov decision processes (POMDPs), enabling principled incorporation of suggestions into agents' belief updates (Asmar & Kochenderfer, 2022). These approaches assume fixed parameters describing suggestion reliability, a significant practical limitation as suggestion quality often varies unpredictably and dynamically.

Other research addresses uncertainty through agent-initiated information gathering. For instance, agents actively query humans to clarify internal state or intent uncertainties (Sadigh et al., 2016; Cui et al., 2023). Ren et al. (2023) recently demonstrated that robots strategically requesting assistance based on uncertainty alignment significantly enhance decision quality and reduce human workload. Earlier work on human–robot communication has also examined when to query humans for input using value-of-information criteria (Kaupp et al., 2010), motivating the need for principled query strategies in collaborative settings. However, existing methods have not yet explicitly integrated dynamic inference of suggestion reliability with proactive suggestion requests.

Recent work has also emphasized two complementary directions. First, trust-aware planning explicitly models and adapts to human trust as a latent variable, showing how calibrated trust can improve team

performance (Chen et al., 2020). Second, language-grounded multi-agent reinforcement learning aligns emergent communication protocols with natural language to enable more interpretable and generalizable interaction (Li et al., 2025). Both directions highlight the importance of trust calibration and interpretable communication in human–agent collaboration. Our focus is orthogonal: rather than modeling trust directly or grounding communication, we treat suggester reliability as a hidden state within the POMDP and provide an explicit ask mechanism for strategic information gathering.

Additionally, prior frameworks generally assume rational or nearly optimal suggesters, a simplification that may not accurately represent practical human-agent interactions. Real-world scenarios frequently involve heuristic, inconsistent, or partially informed suggesters whose guidance does not strictly adhere to rational decision-making models. Thus, there is a clear need for frameworks capable of robustly leveraging diverse suggestion sources, including heuristic or suboptimal guidance.

Our approach also relates to decision-support and explainability in human–robot teaming. Recent work emphasizes justification mechanisms to make autonomous decision support more transparent (Luebbers et al., 2023) and characterizes workload–understanding tradeoffs in explainable AI using an information bottleneck perspective (Sanneman et al., 2024). While complementary, these directions do not embed reliability inference and query management within a sequential decision-making framework.

To address these challenges, we propose a unified POMDP-based framework capable of dynamically inferring suggester reliability and strategically managing information acquisition. Our contributions include: integrating suggester reliability into the agent's belief state, enabling continuous inference and adaptive trust calibration; and introducing an explicit *ask action*, allowing agents to proactively solicit guidance at critical decision points. Experimental evaluations across multiple scenarios demonstrate robust performance, adaptability to dynamically changing suggester reliability, and effective strategic querying behavior. By overcoming previous methodological limitations, our framework advances toward more realistic and practical collaborative scenarios, enhancing the applicability and robustness of autonomous decision-making systems in uncertain environments.

## 2 BACKGROUND

Before presenting our contributions, we briefly review the relevant background on Partially Observable Markov Decision Processes (POMDPs) and Mixed Observable Markov Decision Processes (MOMDPs). These frameworks provide the foundation for our approach to dynamically incorporating suggestion reliability into agent decision-making.

### 2.1 POMDP FORMULATION AND NOTATION

A Partially Observable Markov Decision Process (POMDP) provides a mathematical framework for sequential decision-making under uncertainty where the true system state is not fully observable (Kochenderfer et al., 2022). A POMDP is defined by the tuple $(\mathcal{S}, \mathcal{A}, \mathcal{O}, T, O, R, \gamma)$, where $\mathcal{S}$ is the state space, $\mathcal{A}$ is the action space, and $\mathcal{O}$ is the observation space. The transition function $T(s, a, s')$ defines state transition probabilities, while the observation function $O(o, a, s')$ specifies the probability of receiving observation $o$ after transitioning to state $s'$ with action $a$. The reward function $R(s, a)$ provides immediate rewards, and $\gamma \in [0, 1)$ is the discount factor.

Agents maintain a belief $b \in \mathcal{B}$, representing a probability distribution over possible states. Beliefs are updated using Bayes' rule $b'(s') = \eta O(o, a, s') \sum_{s \in \mathcal{S}} T(s, a, s') b(s)$, where $\eta$ is a normalization constant. A policy $\pi : \mathcal{B} \to \mathcal{A}$ maps beliefs to actions to maximize the expected cumulative discounted reward.

### 2.2 MIXED OBSERVABLE MARKOV DECISION PROCESSES

A Mixed Observable Markov Decision Process (MOMDP) generalizes the POMDP framework by decomposing the state space into fully observable ($\mathcal{X}$) and partially observable ($\mathcal{Y}$) components (Ong et al., 2009; Araya-López et al., 2010). A MOMDP is defined by the tuple $(\mathcal{X}, \mathcal{Y}, \mathcal{A}, \mathcal{O}, T_x, T_y, O, R, \gamma)$, where transitions of observable states are governed by $T_x(x, y, a, x')$ and hidden states by $T_y(x, y, a, x', y')$.

Beliefs in MOMDPs are represented as $(x, b_y)$, where $x \in \mathcal{X}$ is observed directly and $b_y \in \Delta(\mathcal{Y})$ is a belief over hidden states. This factorization significantly reduces computational complexity, enhancing scalability when the fully observable component $|\mathcal{X}|$ is large compared to $|\mathcal{Y}|$. This computational efficiency makes MOMDPs particularly advantageous for modeling scenarios with added complexity, such as incorporating hidden suggester reliability, without incurring intractable computational costs.

# 3 INTEGRATING SUGGESTER TYPES

We consider a sequential decision-making problem under uncertainty modeled as a discrete-state POMDP, involving two entities: an autonomous agent performing actions based on its policy, and an external suggester providing intermittent action recommendations. The suggester observes the environment independently but does not directly alter it, communicating exclusively through action suggestions. Both entities share a common objective of maximizing cumulative discounted reward but maintain separate beliefs due to distinct observations or capabilities.

The agent incorporates suggestions as observations into its belief updates, maintaining autonomy while leveraging external guidance. Crucially, the suggester's reliability is unknown and potentially dynamic, requiring the agent to infer and adapt its level of trust over time. Furthermore, the agent proactively decides when to seek suggestions, balancing informational value against query costs.

## 3.1 MODELING SUGGESTER RELIABILITY

A key limitation of previous approaches, including those described in earlier sections, is the assumption of known and static suggester quality parameters. In practice, suggester reliability may be uncertain or change dynamically. To address this, we explicitly model suggester quality as part of the hidden state that must be inferred during interaction.

The integration of suggester types into the state space can be generalized for any suggestion model $p(\sigma \mid s)$, where $\sigma \in \mathcal{A}$ represents the received suggestion. For clarity, we apply our idea to the noisy rational suggester model (Asmar & Kochenderfer, 2022), defined as $p(\sigma \mid s, \lambda) \propto \exp(\lambda Q(s, \sigma))$, where $Q(s, \sigma)$ denotes the action-value at state $s$ for action $\sigma$, and $\lambda$ is the rationality coefficient characterizing suggester quality.

We discretize suggester quality into a finite set of types $\mathcal{T} = \{\hat{\lambda}_1, \hat{\lambda}_2, \ldots, \hat{\lambda}_m\}$, with each type $\hat{\lambda}_i$ corresponding to a rationality coefficient. The parameter $\lambda$ directly influences suggestion quality: lower values produce nearly random suggestions, while higher values produce deterministic recommendations favoring optimal actions.

For instance, given a state with three actions and action-values $Q(s, a_1) = 5$, $Q(s, a_2) = 4$, and $Q(s, a_3) = 3$, a rationality coefficient $\lambda = 0$ yields equal probabilities $1/3$ for each action. For $\lambda = 1$, probabilities become approximately $\{0.67, 0.24, 0.09\}$, and for $\lambda = 5$, probabilities approach deterministic selection. Discretizing into types such as $\mathcal{T} = \{0, 1, 2, 5, 10\}$ creates a computationally manageable representation spanning random to highly rational suggesters.

The state space of a given problem is expanded to $\mathcal{S} \times \mathcal{T}$. Assuming independence between agent observations and suggestions, belief updates about both environmental states and suggester quality are performed using Bayesian inference:

$$b'(s', \hat{\lambda}') \propto p(o \mid a, s')p(\sigma \mid s', \hat{\lambda}') \sum_{s \in \mathcal{S}, \hat{\lambda} \in \mathcal{T}} p(s', \hat{\lambda}' \mid s, \hat{\lambda}, a)b(s, \hat{\lambda}) \tag{1}$$

where $\hat{\lambda} \in \mathcal{T}$ denotes suggester type, $b$ is the prior belief, $b'$ the updated belief, $o$ the agent observation, and $\sigma_t$ the received suggestion. The independence assumption permits separate modeling of environmental observations and suggestions in belief updates.

While augmenting the state space with suggester types potentially increases computational demands, these can be mitigated using a MOMDP formulation. By placing suggester types in the hidden state component ($\mathcal{Y} \times \mathcal{T}$) and keeping directly observable states in $\mathcal{X}$, computational complexity is reduced, enabling efficient inference while dynamically adapting to suggester reliability.

## 3.2 DYNAMIC SUGGESTERS

The joint transition model, $p(s', \hat{\lambda}' \mid s, \hat{\lambda}, a)$, can often factor into conditionally independent processes $p(s', \hat{\lambda}' \mid s, \hat{\lambda}, a) = p(s' \mid s, a)p(\hat{\lambda}' \mid s, \hat{\lambda}, a)$, separating environmental dynamics and suggester-type transitions. The simplest model assumes static suggester reliability, i.e., $p(\hat{\lambda}' \mid s, \hat{\lambda}, a_t) = 1$ if $\hat{\lambda}' = \hat{\lambda}$.

However, to reflect realistic conditions such as operator fatigue or changing environments, we propose a dynamic suggester model allowing transitions between types with probability $t_p$

$$p(\hat{\lambda}' \mid s, \hat{\lambda}, a) = p(\hat{\lambda}' \mid \hat{\lambda}) = \begin{cases} 1 - t_p & \text{if } \hat{\lambda}' = \hat{\lambda}, \\ \frac{t_p}{|\mathcal{T}| - 1} & \text{otherwise.} \end{cases} \tag{2}$$

Under this model, absent new suggestions, belief distributions over suggester types gradually converge toward uniform uncertainty.

This dynamic model provides adaptability by enabling the agent to continuously reassess suggester reliability, reducing overconfidence and enhancing robustness in changing environments. However, it involves trade-offs; if the true suggester type remains static, beliefs may unnecessarily dilute over time, potentially yielding suboptimal performance compared to a static reliability assumption. When suggester type transitions have known structure, explicitly modeling these dynamics can improve inference precision and overall performance. While our general adaptive model emphasizes robustness to uncertainty, incorporating known or structured dynamics can enhance performance in scenarios where such information is reliably available.

# 4 INCORPORATING AN ASK ACTION

Thus far, our formulation has assumed suggesters autonomously provide recommendations at times of their choosing. Such passive recommendation strategies place timing entirely in the hands of the suggester, who must continuously evaluate when suggestions are beneficial. To grant the agent greater control over information gathering, we introduce an explicit *ask action*. This action allows agents to proactively request suggestions at strategically valuable moments, analogous to sensor queries common in partially observable domains like RockSample (Smith & Simmons, 2004).

## 4.1 DESCRIPTION OF THE ASK ACTION

We define the ask action $a_{\text{ask}} \in \mathcal{A}$ as an explicit information-gathering action whose sole purpose is to elicit a suggestion. The precise dynamics following an ask action depend on the underlying problem structure. For example, in RockSample, executing $a_{\text{ask}}$ leaves the environment state unchanged, allowing the agent to acquire information without altering its position. Conversely, in dynamic scenarios such as the Tag domain (Pineau et al., 2003), the agent performing an ask action remains stationary while external elements, such as target movement, continue evolving.

To implement the ask action, we expand the observation space to include suggestions corresponding to each feasible action (excluding the ask action itself). When the agent executes $a_{\text{ask}}$, it receives a suggestion $\sigma \in \mathcal{A}$ drawn from the suggester model $p(\sigma \mid s, \hat{\lambda})$ and incurs an associated cost. Computing this distribution requires known action-values $Q(s, a)$ for the noisy rational model. We address this through a two-stage approach: first solving the original POMDP without the ask action to derive state-action values, then using these values to parameterize the suggestion observation model $p(\sigma \mid s, \hat{\lambda}) \propto \exp(\hat{\lambda} Q(s, \sigma))$. This bootstrapping method enables the agent to reason effectively about the value of received suggestions within its existing policy framework.

## 4.2 CONSTRAINING SUGGESTION REQUESTS

While the ask action provides strategic flexibility, unrestrained querying can burden suggesters, particularly human collaborators, and potentially degrade the quality of subsequent recommendations. To mitigate this, we propose two complementary constraints.

First, we impose a direct cost on executing the ask action, integrated into the agent's reward function. By appropriately calibrating this cost, the agent is incentivized to request suggestions only

Table 1: Comparison of different agents with different quality of suggesters.

| Agent Type | Tag | | | RS(7, 8, 20, 0) | | | RS(8, 4, 10, −1) | | |
|---|---|---|---|---|---|---|---|---|---|
| | $\lambda^* = 1.0$ | $\lambda^* = 2.0$ | $\lambda^* = 5.0$ | $\lambda^* = 1.0$ | $\lambda^* = 2.0$ | $\lambda^* = 5.0$ | $\lambda^* = 1.0$ | $\lambda^* = 2.0$ | $\lambda^* = 5.0$ |
| Normal | $-10.7 \pm 0.1$ | $-10.7 \pm 0.1$ | $-10.7 \pm 0.1$ | $21.6 \pm 0.1$ | $21.6 \pm 0.1$ | $21.6 \pm 0.1$ | $10.2 \pm 0.1$ | $10.2 \pm 0.1$ | $10.2 \pm 0.1$ |
| Perfect | $-2.3 \pm 0.1$ | $-2.3 \pm 0.1$ | $-2.3 \pm 0.1$ | $28.5 \pm 0.1$ | $28.5 \pm 0.1$ | $28.5 \pm 0.1$ | $16.7 \pm 0.1$ | $16.7 \pm 0.1$ | $16.7 \pm 0.1$ |
| Naive | | | | | | | | | |
| $\nu = 1.00$ | $-13.7 \pm 0.1$ | $-7.7 \pm 0.1$ | $-3.3 \pm 0.1$ | $12.4 \pm 0.1$ | $21.1 \pm 0.1$ | $27.6 \pm 0.1$ | $3.2 \pm 0.1$ | $12.2 \pm 0.1$ | $16.3 \pm 0.1$ |
| $\nu = 0.75$ | $-13.2 \pm 0.1$ | $-8.9 \pm 0.1$ | $-5.0 \pm 0.1$ | $15.2 \pm 0.1$ | $20.8 \pm 0.1$ | $25.2 \pm 0.1$ | $5.2 \pm 0.1$ | $11.2 \pm 0.1$ | $14.4 \pm 0.1$ |
| $\nu = 0.50$ | $-12.7 \pm 0.1$ | $-10.4 \pm 0.2$ | $-8.0 \pm 0.1$ | $17.6 \pm 0.1$ | $20.8 \pm 0.1$ | $23.2 \pm 0.1$ | $7.3 \pm 0.1$ | $10.5 \pm 0.1$ | $12.4 \pm 0.1$ |
| Noisy | | | | | | | | | |
| $\lambda = 5.00$ | $-9.7 \pm 0.1$ | $-5.1 \pm 0.1$ | $\mathbf{-2.8 \pm 0.1}$ | $15.0 \pm 0.1$ | $23.3 \pm 0.1$ | $\mathbf{27.9 \pm 0.1}$ | $10.0 \pm 0.1$ | $14.8 \pm 0.1$ | $\mathbf{16.5 \pm 0.1}$ |
| $\lambda = 2.00$ | $-8.3 \pm 0.1$ | $\mathbf{-4.9 \pm 0.1}$ | $-3.1 \pm 0.1$ | $21.8 \pm 0.1$ | $\mathbf{26.3 \pm 0.1}$ | $27.7 \pm 0.1$ | $12.2 \pm 0.1$ | $\mathbf{15.7 \pm 0.1}$ | $16.1 \pm 0.1$ |
| $\lambda = 1.00$ | $\mathbf{-8.0 \pm 0.1}$ | $-5.7 \pm 0.1$ | $-3.9 \pm 0.1$ | $\mathbf{23.6 \pm 0.1}$ | $25.6 \pm 0.1$ | $26.9 \pm 0.1$ | $\mathbf{13.1 \pm 0.1}$ | $15.2 \pm 0.1$ | $15.3 \pm 0.1$ |
| $\mathcal{T} = \{0, 1, 2, 5, 10\}$ | | | | | | | | | |
| $t_p = 0.00$ | $\mathbf{-8.0 \pm 0.1}$ | $\mathbf{-5.0 \pm 0.1}$ | $\mathbf{-2.8 \pm 0.1}$ | $\mathbf{23.6 \pm 0.1}$ | $\mathbf{26.3 \pm 0.1}$ | $\mathbf{27.8 \pm 0.1}$ | $\mathbf{13.0 \pm 0.1}$ | $\mathbf{15.6 \pm 0.1}$ | $\mathbf{16.5 \pm 0.1}$ |
| $t_p = 0.05$ | $\mathbf{-8.0 \pm 0.1}$ | $\mathbf{-5.0 \pm 0.1}$ | $\mathbf{-2.8 \pm 0.1}$ | $23.3 \pm 0.1$ | $26.0 \pm 0.1$ | $\mathbf{27.8 \pm 0.1}$ | $12.9 \pm 0.1$ | $15.5 \pm 0.1$ | $\mathbf{16.5 \pm 0.1}$ |

when genuinely beneficial. Second, we explicitly limit the total allowable number of ask actions by augmenting the visible state space $\mathcal{X}$ with a discrete counter state. The visible state becomes $\mathcal{X} \times \{0, 1, \ldots, N_{\text{ask}}\}$, with $N_{\text{ask}}$ representing the maximum number of ask actions permitted. Each execution of $a_{\text{ask}}$ decrements this counter, and when it reaches zero, the ask action becomes unavailable, requiring strategic allocation of these limited queries.

Although this expanded state space increases computational requirements, the MOMDP structure places the counter within the fully observable component ($\mathcal{X}$), mitigating computational complexity compared to standard POMDP formulations, as computational cost primarily scales with the hidden state dimension ($\mathcal{Y}$).

## 5 EXPERIMENTAL EVALUATION

To evaluate our proposed integration of suggester types and the ask action, we conducted experiments across the Tag (Pineau et al., 2003) and RockSample (Smith & Simmons, 2004) domains, as previously described in Asmar & Kochenderfer (2022). Specifically, we used the slightly modified transition dynamics for Tag, making the scenario marginally more challenging, and the standard RockSample(7,8) problem without sensor costs and RockSample(8,4) with a sensor cost of $-1$. Both domains were formulated as MOMDPs, enabling efficient belief maintenance and dynamic adaptation to varying suggester quality. Policies were computed using the SARSOP algorithm (Kurniawati et al., 2008) via the POMDPs.jl framework (Egorov et al., 2017). Experiments were conducted on a MacBook Pro with an Apple M1 Max processor and 32 GB of memory.

To analyze belief adaptation and decision-making over extended periods, we employed a repeated-reset approach rather than traditional terminal conditions. In RockSample, upon reaching the exit, rock samples were reinitialized randomly, and the agent returned to its initial position. Similarly, in Tag, after each successful tag, the agent and opponent were repositioned randomly without overlap.

Each simulation comprised multiple trials, where a trial lasted from initialization until a successful tag (Tag) or environment exit (RockSample). Numerous simulations were conducted to ensure statistical robustness. Unless noted otherwise, metrics are reported as per-trial averages across all trials and simulations. For dynamic suggester evaluations, we also report trial-by-trial averages to highlight temporal adaptation. All results include $95\%$ confidence intervals.

### 5.1 PERFORMANCE WITH STATIC SUGGESTER TYPES

We first evaluated agent performance under static suggester conditions, employing noisy rational suggesters characterized by various rationality coefficients ($\lambda$), with the true suggester rationality denoted as $\lambda^*$. Agents received suggestions at every time step, updating their beliefs according to the model described in Section 3 prior to action selection.

Quantitative results are summarized in Table 1. Consistent with prior findings, naive agents (those not modeling suggester reliability) exhibited performance heavily dependent on suggester quality: higher rationality coefficients led to improved performance, whereas lower-quality suggestions degraded outcomes significantly. Agents explicitly modeling the correct rationality coefficient ($\lambda^*$) naturally

achieved optimal performance, while discrepancies between assumed and actual suggester quality notably reduced performance.

We further tested agents capable of maintaining beliefs over multiple discrete suggester rationalities, specifically $\mathcal{T} = \{0, 1, 2, 5, 10\}$. Evaluations considered two types of suggester type transition models: static scenarios, where agents assumed no transitions in suggester type; and dynamic scenarios, where transitions followed the dynamics described by eq. (2) with transition probability parameter $t_p$. Results demonstrated that agents maintaining beliefs over multiple suggester types consistently achieved performance comparable to agents with accurate knowledge of the true suggester quality. These outcomes highlight the adaptive robustness conferred by explicitly modeling multiple suggester types.

## 5.2 PERFORMANCE WITH DYNAMIC SUGGESTERS

We next examined agent performance under dynamically changing suggester quality conditions. Specifically, the true suggester rationality coefficient ($\lambda^*$) varied systematically over multiple trials within simulations, transitioning between distinct rationality levels at predetermined intervals. Agent performance, depicted in Figure 1a, shows mean rewards per trial averaged across multiple simulations in the Tag domain. Vertical dashed lines mark transitions in $\lambda^*$, labeled explicitly for clarity. Baseline results from perfect, normal, and fixed-rationality agents (Noisy $\lambda = 1$ and Noisy $\lambda = 5$) are provided for reference.

Fixed-type noisy agents performed predictably: the Noisy $\lambda = 5$ agent excelled during periods of high-quality suggestions ($\lambda^* \geq 2$) but significantly underperformed when suggestions were poor ($\lambda^* = 0, 1$). Conversely, the Noisy $\lambda = 1$ agent, less reliant on suggestions, exhibited more stable performance overall. Multiple-type (MT) agents, capable of adapting beliefs regarding suggester quality, demonstrated consistent robustness across varying conditions.

Two MT agents were evaluated: MT $t_p = 0.00$ (static hypothesis scenario) and MT $t_p = 0.05$ (dynamic hypothesis scenario), both initialized with identical belief distributions over suggester types ($0, 1, 2, 5, 10$ with probabilities $[0.1, 0.2, 0.4, 0.2, 0.1]$). Initially comparable, their performances diverged after transitions to lower-quality suggestions. The dynamic MT agent ($t_p = 0.05$) rapidly adjusted its beliefs, recovering near-optimal performance despite persistently poor suggestions. In contrast, the static MT agent ($t_p = 0.00$) adapted more slowly, resulting in prolonged suboptimal outcomes.

The agents' adaptability is further illustrated in fig. 1b, showing the mean expected suggester type over trials, averaged across simulations. While these averages condense the full distribution into a single value and do not reflect variance explicitly, they clearly indicate the speed and effectiveness of belief adjustments following transitions in suggester quality. The dynamic MT agent ($t_p = 0.05$) consistently adapted its expectations more promptly than the static MT agent ($t_p = 0.00$), underscoring the practical advantage of explicitly modeling dynamic transitions between suggester types.

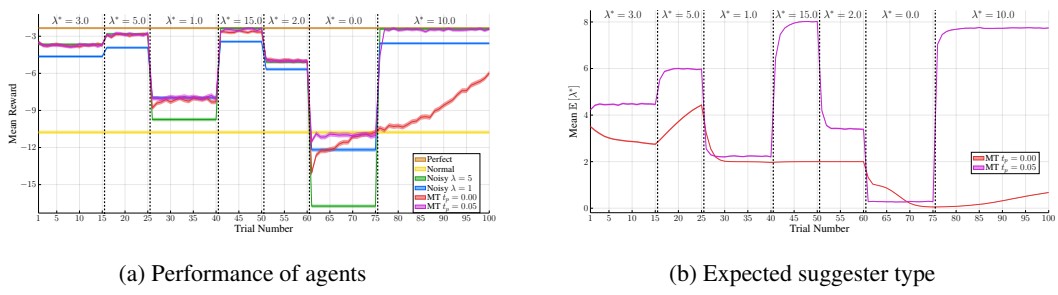

(a) Performance of agents          (b) Expected suggester type

Figure 1: Dynamic suggester evaluation in the Tag domain. Subfigure (a) shows the performance of agents with a dynamic suggester, and subfigure (b) shows the expected suggester type for static and dynamic MT agents across trials. Dashed lines indicate transitions in true suggester rationality ($\lambda^*$).

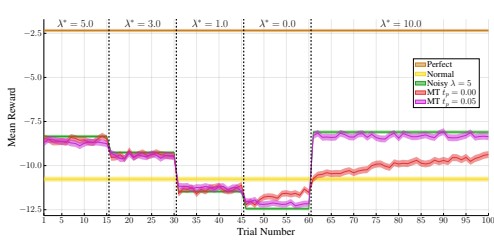
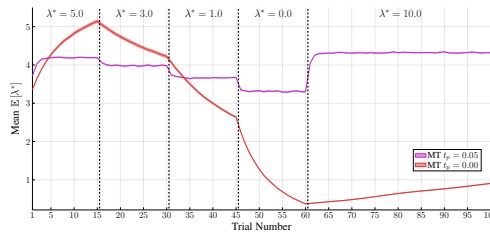

(a) Performance of agents.

(b) Expected suggester type.

Figure 2: Dynamic suggester evaluation in the Tag domain under a constraint of one ask action per trial. Subfigure (a) shows the performance of agents, and subfigure (b) shows the expected suggester type for MT agents.

Table 2: Performance on the Tag domain with unlimited number of asks.

| Metric | Agent Type | $\lambda^* = 0$ | $\lambda^* = 1$ | $\lambda^* = 2$ | $\lambda^* = 5$ | $\lambda^* = 10$ |
|---|---|---|---|---|---|---|
| Reward per Trial | Normal | $-10.77 \pm 0.02$ | $-10.77 \pm 0.02$ | $-10.77 \pm 0.02$ | $-10.77 \pm 0.02$ | $-10.77 \pm 0.02$ |
| | Noisy $\lambda = 1$ | $-11.95 \pm 0.03$ | $-10.77 \pm 0.04$ | $-9.94 \pm 0.04$ | $-9.16 \pm 0.04$ | $-9.04 \pm 0.04$ |
| | Noisy $\lambda = 2$ | $-14.81 \pm 0.03$ | $-11.20 \pm 0.03$ | $-9.09 \pm 0.03$ | $-7.53 \pm 0.03$ | $-7.40 \pm 0.03$ |
| | Noisy $\lambda = 5$ | $-15.01 \pm 0.03$ | $-11.74 \pm 0.04$ | $-9.41 \pm 0.03$ | $-7.34 \pm 0.03$ | $-7.05 \pm 0.03$ |
| | MT $t_p = 0.00$ | $-11.28 \pm 0.04$ | $-11.01 \pm 0.03$ | $-9.13 \pm 0.03$ | $-7.36 \pm 0.03$ | $-7.06 \pm 0.03$ |
| | MT $t_p = 0.05$ | $-12.64 \pm 0.03$ | $-11.09 \pm 0.03$ | $-9.15 \pm 0.03$ | $-7.41 \pm 0.03$ | $-7.17 \pm 0.03$ |
| Asks per Trial | Normal | $0.00 \pm 0.00$ | $0.00 \pm 0.00$ | $0.00 \pm 0.00$ | $0.00 \pm 0.00$ | $0.00 \pm 0.00$ |
| | Noisy $\lambda = 1$ | $2.25 \pm 0.01$ | $1.70 \pm 0.01$ | $1.51 \pm 0.01$ | $1.27 \pm 0.01$ | $1.21 \pm 0.01$ |
| | Noisy $\lambda = 2$ | $20.26 \pm 0.10$ | $7.77 \pm 0.04$ | $4.92 \pm 0.02$ | $3.09 \pm 0.01$ | $2.85 \pm 0.01$ |
| | Noisy $\lambda = 5$ | $21.57 \pm 0.11$ | $9.17 \pm 0.04$ | $5.73 \pm 0.03$ | $3.53 \pm 0.01$ | $3.19 \pm 0.01$ |
| | MT $t_p = 0.00$ | $0.33 \pm 0.01$ | $5.29 \pm 0.02$ | $4.53 \pm 0.02$ | $3.22 \pm 0.01$ | $3.01 \pm 0.01$ |
| | MT $t_p = 0.05$ | $5.75 \pm 0.03$ | $6.37 \pm 0.03$ | $4.50 \pm 0.02$ | $3.04 \pm 0.01$ | $2.80 \pm 0.01$ |
| Steps per Trial | Normal | $35.48 \pm 0.08$ | $35.48 \pm 0.08$ | $35.48 \pm 0.08$ | $35.48 \pm 0.08$ | $35.48 \pm 0.08$ |
| | Noisy $\lambda = 1$ | $39.53 \pm 0.16$ | $35.49 \pm 0.15$ | $32.83 \pm 0.14$ | $30.59 \pm 0.14$ | $30.24 \pm 0.14$ |
| | Noisy $\lambda = 2$ | $80.39 \pm 0.40$ | $37.38 \pm 0.16$ | $27.20 \pm 0.11$ | $22.44 \pm 0.08$ | $22.14 \pm 0.08$ |
| | Noisy $\lambda = 5$ | $85.41 \pm 0.43$ | $42.30 \pm 0.19$ | $29.00 \pm 0.11$ | $21.95 \pm 0.08$ | $21.17 \pm 0.08$ |
| | MT $t_p = 0.00$ | $37.83 \pm 0.16$ | $36.44 \pm 0.15$ | $27.74 \pm 0.11$ | $22.04 \pm 0.08$ | $21.24 \pm 0.08$ |
| | MT $t_p = 0.05$ | $46.65 \pm 0.20$ | $36.93 \pm 0.16$ | $27.96 \pm 0.11$ | $22.33 \pm 0.08$ | $21.71 \pm 0.08$ |

## 5.3 ADAPTIVE USE AND CONSTRAINTS OF THE ASK ACTION

We next investigated the effectiveness of integrating an explicit ask action, enabling agents to strategically request suggestions. Initially, we evaluated performance in the Tag domain without constraints on suggestion requests. Results summarized in Table 2 include metrics such as average reward per trial, steps per trial, and ask actions executed per trial, computed over $10,000$ simulations consisting of 15 trials each.

These results indicate that multiple-type (MT) agents adaptively manage their suggestion requests based on inferred suggester quality. MT agents effectively balanced the benefits of acquiring informative suggestions against the costs of unnecessary requests, closely approaching the performance of agents with correct, known rationality coefficients. A clear illustration is the MT $t_p = 0.00$ agent interacting with a random suggester ($\lambda^* = 0.0$), rapidly recognizing low suggestion quality and reducing ask actions to an average of $0.33 \pm 0.01$ per trial, primarily querying only under initial uncertainty.

In RockSample, we introduced a suggestion request cost equal to the standard sensor usage cost ($-1.0$ per request). Table 3 shows the average number of ask actions per trial for various suggester qualities. Agents typically refrained from requesting suggestions from low-quality suggesters ($\lambda < 2$) due to insufficient informational value relative to the incurred cost. Even MT agents avoided querying lower-quality suggesters unless their initial belief heavily favored higher-quality suggesters. This rational behavior, highlighted in Table 4, underscores the agents' strategic consideration of the informational value against incurred costs.

Table 3: Number of asks per trial on RockSample(8,4).

| Agent | $\lambda^* = 0$ | $\lambda^* = 1$ | $\lambda^* = 2$ | $\lambda^* = 5$ | $\lambda^* = 10$ |
|---|---|---|---|---|---|
| Noisy $\lambda = 1$ | $0.00 \pm 0.00$ | $0.00 \pm 0.00$ | $0.00 \pm 0.00$ | $0.00 \pm 0.00$ | $0.00 \pm 0.00$ |
| Noisy $\lambda = 2$ | $1.72 \pm 0.02$ | $1.33 \pm 0.01$ | $1.06 \pm 0.01$ | $1.00 \pm 0.00$ | $1.00 \pm 0.00$ |
| Noisy $\lambda = 5$ | $2.54 \pm 0.01$ | $2.74 \pm 0.02$ | $2.78 \pm 0.02$ | $2.70 \pm 0.01$ | $2.65 \pm 0.01$ |
| MT $t_p = 0.00$ | $0.00 \pm 0.00$ | $0.00 \pm 0.00$ | $0.00 \pm 0.00$ | $0.00 \pm 0.00$ | $0.00 \pm 0.00$ |
| MT $t_p = 0.05$ | $0.00 \pm 0.00$ | $0.00 \pm 0.00$ | $0.00 \pm 0.00$ | $0.00 \pm 0.00$ | $0.00 \pm 0.00$ |

Table 4: Reward per trial for selected agents on RockSample(8,4) with an unlimited number of asks.

| Agent | $\lambda^* = 0$ | $\lambda^* = 1$ | $\lambda^* = 2$ | $\lambda^* = 5$ | $\lambda^* = 10$ |
|---|---|---|---|---|---|
| Normal | $10.2 \pm 0.1$ | $10.2 \pm 0.1$ | $10.2 \pm 0.1$ | $10.2 \pm 0.1$ | $10.2 \pm 0.1$ |
| Noisy $\lambda = 2$ | $4.9 \pm 0.1$ | $8.9 \pm 0.1$ | $10.4 \pm 0.1$ | $10.8 \pm 0.1$ | $10.9 \pm 0.1$ |
| Noisy $\lambda = 5$ | $3.2 \pm 0.1$ | $6.5 \pm 0.1$ | $9.5 \pm 0.1$ | $11.5 \pm 0.1$ | $11.8 \pm 0.1$ |

We further evaluated performance under constraints limiting the number of ask actions. With only one allowed ask action per trial, agents achieved modest performance gains over the baseline but remained below optimal. We examined a dynamic scenario with sequential transitions in suggester quality ($\lambda^* = 5.0, 3.0, 1.0, 0.0, 10.0$) at predetermined intervals (fig. 2a).

This scenario highlighted the adaptive strengths of MT agents relative to fixed-type agents under strict information constraints. Initially, MT agents matched the performance of the Noisy $\lambda = 5$ agent during periods of high-quality suggestions. However, as suggester quality declined, MT agents adapted better, though performance still fell below that of the Normal agent. The dynamic MT agent ($t_p = 0.05$) adjusted beliefs more quickly, outperforming the fixed-type agent, while the static MT agent ($t_p = 0.00$) improved more slowly over time.

The evolution of mean expected suggester type across trials (fig. 2b) provided further insight into adaptability. This metric, computed as the average expectation of suggester types at the end of each trial, revealed that the static MT agent initially approached accurate beliefs but adapted slowly after quality transitions, gradually improving from trial 60 onward. Conversely, the dynamic MT agent ($t_p = 0.05$), due to modeling uniform suggester transitions, maintained beliefs closer to the stationary distribution mean (3.6), showing swift adaptability in performance but dampened belief precision.

This observation aligns with theoretical mixing time concepts commonly studied in discrete Markov chains (Levin & Peres, 2017). Mixing time quantifies how quickly a Markov process approaches its stationary distribution from an arbitrary initial distribution. Given our transition probability $t_p = 0.05$, the mixing time to reach a total variation distance below $0.1$ from the uniform stationary distribution was approximately 33 steps, closely matching the average trial duration of 32 steps. This rapid convergence explains the dampened belief precision observed for the dynamic MT agent. Employing lower transition probabilities or more structured transition models could mitigate this rapid convergence, enabling more precise belief maintenance and further improving adaptive performance.

## 5.4 EVALUATION WITH A HEURISTIC-BASED SUGGESTER

The previous experiments employed suggesters explicitly modeled using the noisy rational framework with varying rationality coefficients ($\lambda$). To further assess our approach's generalizability and robustness, we evaluated agent performance using a heuristic-based suggester whose suggestion generation mechanism diverged from our noisy rational assumptions.

We implemented a heuristic suggester in the Tag domain, simulating an external sensor with limited sensing capability that is unavailable directly to the agent. Specifically, the heuristic provided directional suggestions (north, west, or east) only when the target was within two grid cells of the corresponding wall and the agent was positioned outside that region. When these conditions were unmet, no suggestions were provided. This scenario realistically mirrors situations in which human operators or external systems offer partial and localized guidance without complete situational awareness.

Table 5: Performance with heuristic-based suggestions on the Tag domain.

| Agent Type | Reward per Trial | Asks per Trial |
|---|---|---|
| Noisy $\lambda = 5$ | $-7.53 \pm 0.05$ | $3.72 \pm 0.03$ |
| Noisy $\lambda = 1$ | $-9.11 \pm 0.06$ | $1.11 \pm 0.01$ |
| MT $t_p = 0.00$ | $-7.46 \pm 0.05$ | $3.75 \pm 0.03$ |
| MT $t_p = 0.05$ | $-7.35 \pm 0.05$ | $3.10 \pm 0.02$ |

Results summarized in Table 5 indicate that incorporating heuristic-based suggestions within our noisy rational modeling framework significantly improved agent performance compared to scenarios lacking suggestions. This improvement underscores our framework's ability to effectively leverage information from heuristic suggestions, despite deviations from assumed rationality. The MT agent employing dynamic type transitions ($t_p = 0.05$) effectively balanced suggestion requests, achieving superior performance with fewer average asks per trial compared to other agents. This outcome highlights the MT framework's adaptability, successfully interpreting heuristic suggestions without explicit alignment with their underlying generative mechanism.

A naive agent dependent solely on heuristic suggestions would struggle due to insufficient directional precision. However, our integrated approach robustly leverages partial, noisy guidance, demonstrating considerable flexibility and robustness beyond the originally modeled noisy rational paradigm.

# 6 DISCUSSION

Effectively integrating external suggestions into autonomous decision-making remains a critical challenge, especially under uncertainty and dynamically varying conditions. Our experiments show that agents capable of maintaining multiple hypotheses regarding suggester reliability consistently demonstrate greater adaptability across static and dynamic scenarios than those relying on fixed assumptions. Explicitly modeling transitions between suggester types enhances the speed and accuracy of belief recalibration, particularly valuable in uncertain or changing environments. Incorporating the ask action further improves decision-making efficiency, allowing agents to selectively query suggesters only when anticipated informational gains outweigh the costs.

These findings offer considerable practical implications, especially for real-world applications involving human-agent collaboration, multi-agent coordination, or networks with fluctuating reliability. The adaptive framework developed here helps autonomous systems better manage uncertainty, reduces their dependence on consistently reliable external guidance, and effectively leverages partial or intermittent information. Consequently, this approach enhances system robustness and decision-making quality in dynamic and uncertain environments commonly encountered in practical scenarios.

Nonetheless, our approach has several limitations related primarily to modeling assumptions. While discretizing suggester reliability into finite types provided computational convenience and efficient inference, the noisy rational model may not capture real-world suggestion processes. Our chosen set of rationality coefficients ($\lambda$) spanned from completely random to highly rational based on the computed policy, which, although practical for evaluation, might not accurately reflect real-world suggester behavior. Although our heuristic-based suggester experiments demonstrated promising robustness, additional research is needed to evaluate our approach with a broader range of non-rational or heuristic suggestion mechanisms, including scenarios with explicit deviations from modeled rationality assumptions.

Furthermore, the simplified uniform transition model employed for suggester reliability dynamics may insufficiently reflect practical conditions, where changes in reliability often correlate with specific identifiable events, such as environmental shifts, hardware modifications, or operator changes. Future work could enhance our framework by explicitly associating suggester type transitions with such events. Developing event-driven, context-sensitive models for suggester reliability would likely improve belief accuracy, decision-making performance, and overall flexibility.

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

## A  LLM USAGE

Large language models (LLMs) were used in a limited, assistive capacity during this project. Specifically:

- Writing and editing: LLMs were used to help refine wording, improve clarity, and reduce redundancy in the paper's text. All technical content, modeling decisions, and results are original and authored by the listed authors.
- Coding assistance: LLMs were occasionally used to help debug implementation issues. The modeling choices, algorithms, and experimental design were fully specified and implemented by the authors.

No part of the research ideation, methodology, or results was generated by an LLM. The authors take full responsibility for the content of this paper.

## B  REPRODUCIBILITY STATEMENT

Our experiments use standard benchmark domains (Tag and RockSample) with clearly specified modifications (e.g., enabling the ask action, modeling suggester types, and defining priors). Policies were computed with SARSOP via POMDPs.jl, and all solver settings, domain variants, and simulation functions are included in the provided code. The supplementary materials contain the complete repository with setup scripts, policy generators, and simulation functions for reproducing data for the tables and figures in the paper. The accompanying README outlines the code structure and provides detailed instructions on running the functions to perform the experiments. After review, we will release this repository publicly to further support transparency and reproducibility.

