# OpenReview forum: "Learning to Trust: Bayesian Adaptation to Varying Suggester Reliability in Sequential Decision Making"
_ICLR.cc/2026/Conference — Submitted to ICLR 2026_

### Official Review · Reviewer_Qnk9 · 2025-10-31

**Soundness:** 3
**Presentation:** 3
**Contribution:** 3
**Rating:** 4
**Confidence:** 3

**Summary:**

The paper proposes a POMDP-based framework that allows an autonomous agent to (1) maintain a Bayesian belief over discrete “suggester types” that encode unknown and possibly time-varying reliability, and (2) actively request suggestions via an explicit, cost-bearing “ask” action. Belief updates are performed with a factored MOMDP representation to keep computation tractable. Extensive experiments on Tag and RockSample show that the agent quickly adapts its trust when suggester quality drifts, and strategically limits costly queries. A final heuristic-suggester ablation demonstrates robustness to model mismatch.

**Strengths:**

Novel integration of latent suggester reliability and agent-initiated queries inside a single Bayesian decision-theoretic framework.
Sound modeling choice: MOMDP factorization keeps the hidden state small, enabling off-the-shelf solvers (SARSOP) to scale to the augmented state space.

**Weaknesses:**

Discrete-type assumption: real-world reliability is almost certainly continuous and context-dependent; the chosen five-point discretization may be too coarse and is not motivated by data.
Scalability concerns: experiments are limited to small toy domains; the hidden component Y×T is still |T| times larger, which will hurt solvers when |S| or the horizon grows.
Limited novelty in ask mechanism: “query=information-gathering action” is well-known in POMDP sensor management; the paper does not theoretically analyze value-of-information or provide new solver tricks.

**Questions:**

1. How sensitive are the policies to the granularity of T and to the exact numeric ask-cost used? Any theoretical analysis?
2. Experiments are limited to small toy domains, which is not convincing. Please outline how the same POMDP formulation would be instantiated when the action space is continuous (e.g., 2-D mouse drag, 6-DOF robot joint commands) and suggestions arrive as natural-language or GUI-event streams. Would you discretize the continuous space, or move to a continuous-state POMDP / POMDP-lite solver?

---

> ### Author Response · Authors · 2025-11-22
>
> Thank you for the thoughtful review and for highlighting the strengths of the modeling formulation. We address your concerns and questions below.
>
> **Discrete-type assumption**
>
> We agree that real-world reliability is often continuous and context dependent. We chose a discrete representation because it allows exact Bayesian filtering within an MOMDP and supports the use of efficient offline POMDP solvers such as SARSOP. Moving to continuous reliability would require approximate belief representations (e.g., particle filtering) and continuous-state POMDP solvers, which introduce additional complexity. Empirically, the exact λ values are less important than spanning the behavioral range from low to high reliability. For example, λ values of 3.0 and 3.5 produce nearly identical suggestion distributions, while larger differences such as λ = 0 versus λ = 10 matter significantly. Even when the true λ lies between grid values, the belief converges toward the appropriate region of the latent space, which reflects a natural benefit of reasoning over uncertainty in a POMDP framework.
>
> **Scalability concerns**
>
> We agree that POMDPs are difficult to scale. Our contribution is primarily a modeling one that shows how to jointly infer suggester quality and environment state using a tractable latent structure. As POMDP solvers continue to improve, the technique will scale naturally with them. This structure is compatible with online and approximate solvers for larger or more realistic domains.
>
> **Novelty of the ask mechanism**
>
> We agree that information-gathering actions are well studied in POMDPs and that this is the key motivation for including the ask action. The contribution here is showing how such an action interacts with a reliability-latent-variable model. When the agent can infer suggester quality, the ask action becomes a tool that is used selectively when suggestions are valuable and avoided when not. In fact, examining the resulting policies often highlights critical decision points. For example, in RockSample(8,4), the optimal time to ask frequently occurred near the middle of the map, where the agent must choose between diverging paths. Although we explored value-of-information reasoning early in development, we found that the optimal policies already captured these insights, and visualizing the experiments illustrate this behavior.
>
> **Responses to Reviewer Questions**
>
> *Q1. Sensitivity to granularity of T and to the numeric ask-cost. Any theoretical analysis?*
>
> We do not have formal theoretical guarantees, but we can comment on empirical behavior. In our experiments, performance was not sensitive to fine-grained choices of λ. Because λ enters as a scaling factor in a softmax over action values, small numerical differences have little effect on the suggestion distribution. What mattered was covering the meaningful behavioral span from near-random suggestion (low λ) to near-policy-aligned suggestion (high λ). Even when the true λ was not in the discrete set, the belief tended to converge toward a weighted region of types that reflected actual suggester behavior. This is a natural strength of performing inference in a POMDP.
>
> Ask-cost sensitivity behaves as expected: different domains have different critical decision points where asking becomes valuable. We varied ask costs modestly and saw consistent qualitative behavior. A broader exploration of ask-cost tuning is a natural direction for future work, but our goal here was to study the modeling idea rather than perform a cost-sensitivity sweep.
>
> *Q2. How does the formulation extend to continuous action spaces or natural-language / GUI-based suggestion channels?*
> This connects to a broader challenge in scaling POMDP models to complex real-world domains. In practice, POMDPs should be used at an abstraction level that captures the decision structure while keeping the state and action spaces manageable. For continuous state or action spaces (e.g., 2-D drags or 6-DOF motion), one would typically use a continuous-state POMDP solver or an approximate online method. Our framework only requires a distribution over suggester outputs conditioned on the underlying state or belief. If natural-language or GUI-event suggestions can be mapped to a distribution over abstract actions or action categories, they can be incorporated into the suggestion observation model directly. The mechanism for obtaining that distribution is problem dependent, but the belief-update machinery remains the same. Any solver capable of maintaining beliefs and producing policies in the chosen abstraction (e.g., continuous-state POMDPs or POMDP-lite methods) could be used.

---

### Official Review · Reviewer_ZMv2 · 2025-10-31

**Soundness:** 2
**Presentation:** 2
**Contribution:** 2
**Rating:** 2
**Confidence:** 3

**Summary:**

The paper considers integrating external suggestions (e.g. from humans) into autonomous decision-making. In this context, the paper proposes modelling (i) suggester’s suggestion to be distributed as a tempered action-value function with a (temperature) rationality parameter $\lambda \geq 0$, (ii) dynamic suggester type (that is characterized by discretized $\lambda$) as a lazy random walk, and (iii) incorporating “ask action” option to request suggestion.

**Strengths:**

The method section 3 is easy to follow and the proposed contributions/components are introduced clearly with motivations. The paper also experiments in the setting where the proposed suggester model is misspecified (Section 5.4).

**Weaknesses:**

The contributions/components (i)–(iii) listed in the summary box are somewhat orthogonal, especially (i)–(ii) relative to (iii). Without comprehensive empirical experiments demonstrating a significant performance improvement over justified baselines, the overall contribution looks like a sum of incremental components. Further, proper ablation studies are critical in this case to understand the strengths and weaknesses of the individual components (maybe Tables 1–2 may touch on this, but it is difficult to discern without a clear narrative thread in the main text).

Experimental Section 5 is insufficient: it lacks proper discussion of the hypothesis, baselines, and evaluation metrics. For this reason, it is difficult to judge (i) whether the proposed method preforms well overall, (ii) what are the components of the proposed method that contribute the most, i.e. ablation studies, and (iii) what is the trade-off between the increased computational complexity and the improvement in empirical performance.

“Results summarized in Table 5 indicate that incorporating heuristic-based suggestions within our noisy rational modeling framework significantly improved agent performance compared to scenarios lacking suggestions.” Without proper discussion of the experimental setup conclusions like that are difficult to judge.

“Numerous simulations were conducted to ensure statistical robustness.” This is too vague, etc.

**Questions:**

As $\lambda$ is continuous, why not treat is as such rather than discretize? Is there some other reason to keep it discrete than convenience?

“…We address this through a two-stage approach: first solving the original POMDP without the ask action to derive state-action values, then using these values to parameterize the suggestion observation model…” Sounds computationally heavy, does it?

---

> ### Author Response · Authors · 2025-11-22
>
> Thank you for the thoughtful review. We are glad the modeling components were found clear and well motivated. We address your concerns and questions below.
>
> **On the relationship between components (i)–(iii):** The three components address different but complementary aspects of suggestion-based decision making.
> (i) Modeling suggester reliability as a latent variable is the core contribution. It enables joint inference over the environment state and suggester quality, which prior fixed-reliability approaches cannot support.
> (ii) Dynamic suggester types extend this to settings where human or sensor reliability varies over time, which is common in practice.
> (iii) The ask action leverages the reliability inference mechanism. It allows the agent to query strategically when it believes the suggester is valuable, and to avoid unnecessary or low-value queries.
>
> Thus, while orthogonal in implementation, the components address different but complementary challenges. The ablations reflect this separation:
> - Table 1 isolates (i)
> - Table 2 compares static and dynamic reliability (i) vs (ii)
> - Tables 3 and 5 evaluate querying behavior (iii)
> - Cross-table comparisons illustrate how the components interact.
>
> **On experimental discussion and baselines:** In planning-oriented POMDP work, expected discounted reward is the standard evaluation metric because it is exactly what solvers optimize. We therefore focus on this metric. To our knowledge, there are no prior baselines for adaptive trust over suggestion reliability in partially observable environments beyond fixed-λ approaches. Our comparison set therefore includes the relevant established alternatives:
> – an agent without suggestions,
> – fixed-λ agents,
> – multi-type agents under both static and dynamic reliability models.
>
> Regarding computational tradeoffs, the additional latent variable increases complexity as the number of types increases. We can compare the computational cost tradeoff by comparing results with static types and those with multiple types.
>
> **On interpreting the heuristic-suggester experiments:** The improvement over the no-suggestion baseline is substantial in Tag (e.g. −10.8 in Table 3 vs higher rewards in Table 5). The purpose of this experiment is to show that the agent can benefit from suggestions even when the suggester does not follow the assumed noisy-rational model. We can expand the narrative in the final version for clarity if needed and desired.
>
> **On “numerous simulations”:** We will include explicit counts in the final version if desired. The aim was to emphasize that statistically stable averages were obtained, but we agree that including the numbers is clearer.
>
> **Responses to Reviewer Questions**
>
> *Q1. As λ is continuous, why discretize it? Is there any reason beyond convenience?*
>
> Yes. Discretization enables exact Bayesian filtering in the MOMDP and allows us to use efficient offline solvers like SARSOP. Treating λ as continuous would require continuous-state POMDP machinery such as particle filters or other approximate belief representations, which introduces additional solver complexity and reduces comparability across experiments. Our goal here was to isolate the value of jointly inferring reliability and the state, so we chose a representation that supports exact inference and a static computed policy.
>
> Empirically, covering the behavioral span (poor, moderate, strong suggesters) mattered far more than fine granularity. For example, λ values of 3 and 3.5 produce nearly identical suggestion distributions. The span of types, not discretization density, drove performance.
>
> *Q2. Does the two-stage approach (solving the POMDP first to obtain Q-values) introduce heavy computation?*
>
> Solving a POMDP is inherently computationally hard, but this is standard in planning under uncertainty and does not introduce extra burden beyond typical practice. The two-stage structure follows common shared-autonomy approaches: first compute a base policy, then reason about suggestions relative to that policy. We use the base POMDP solution to obtain action values that parameterize different suggester models, not because the framework requires a fully solved POMDP to operate. Any suggester model would work, we just used suggester models derived from a POMDP solution. Advances in POMDP solvers continue to reduce this cost, and our formulation benefits directly from these improvements.

---

### Official Review · Reviewer_vhfw · 2025-10-31

**Soundness:** 3
**Presentation:** 3
**Contribution:** 2
**Rating:** 4
**Confidence:** 3

**Summary:**

This paper studies autonomous agents operating in POMDPs who receive external action suggestions (e.g., from a human or another agent) whose reliability may vary over time. Prior work typically assumes fixed and known suggester reliability, which does not reflect real human behavior or real-world sensing systems. Main contributions are:
- Model suggester reliability as a latent variable and infer it dynamically via Bayesian updates.
- Introduce an explicit ask action, enabling strategic querying of suggestions under cost.
- Demonstrate robust adaptation to varying suggestion quality and ability to avoid low-value queries.
- Show empirical results across Tag and RockSample, with both rational and heuristic suggesters.

**Strengths:**

- Novel Motivation:
(1) Addresses a real and growing need in human-AI teaming: adapting trust to variable advice quality.
(2) Aligned with the trend of interactive assistance and trust calibration.

- Formulation:
POMDP and MOMDP are modeled in the scenarios: (1) Present a solid use of the MOMDP structure to efficiently manage the expanded state space introduced by modeling suggester reliability as a latent variable. (2) The Bayesian update mechanism for jointly inferring environment state and suggester quality is principled and well motivated.

- Experiments:
(1) The study covers a broad range of settings, including static, dynamic, and heuristic suggesters, as well as scenarios involving ask costs and limitations on querying. (2) It further provides informative ablations, such as fixed-λ models, discrete-type inference, and dynamic type transitions, helping isolate the contributions of each component. (3) Include relevant baseline comparisons: normal agents, naive fixed-λ agents, noisy-rational suggesters, and multi-type agents in both static and dynamic configurations

**Weaknesses:**

- Human study missing: For human-trust motivation, no human-in-the-loop experiments are conducted. Although this paper acknowledges this, it is still important for this paper.

- Scalability: Tag and RockSample are standard but small. What if (1) the larger POMDP domains, (2) higher-dimensional latent human models, (3) multiple suggesters or groups of helpers.

- Reliance on **known** Q-values: The ask suggestion model uses pre-solved Q values. What if (1) Q is inaccurate, (2) Q value needs to be learnt. Is it possible to apply to RL or online learning settings.


- "Does your AI agent get you? A personalizable framework for approximating human models from argumentation-based dialogue traces". This paper seems also estimating the belief.

**Questions:**

Your method discretizes suggester rationality (λ) into a small fixed set:
- Q1: How sensitive is performance to the choice of λ grid values (e.g., {0,1,2,5,10})?

- Q2: Would adaptive or continuous inference over λ (e.g., particle filtering or Bayesian regression) further improve performance?

- Q3: If λ lies between grid points, how does belief estimation degrade?


The ask-action model assumes access to accurate Q-values from the solved base POMDP.

- Q4: How robust is the ask mechanism when Q-values are approximate or learned online (e.g., under model mismatch or RL)?

- Q5: Could errors in estimated Q(s,a) lead to biased belief updates about suggester reliability?

Experiments: Tag and RockSample.

- Q6: How does the computational cost scale with the number of suggester types and belief complexity?

- Q7: Could the method scale to larger, continuous-state problems or multi-human settings?

Human-in-the-loop:
- Q8: Do you anticipate additional challenges when suggesters are real humans? How would you integrate explicit human feedback or confidence signals

---

> ### Author Response · Authors · 2025-11-22
>
> Thank you for the detailed and constructive feedback. We are glad the motivation, formulation, and evaluation were found meaningful. We address the concerns and questions below.
>
> **Human study:** We agree that human studies are valuable. Our goal was first to establish whether modeling suggester reliability as a latent variable and performing Bayesian adaptation is effective in controlled sequential settings. This provides a foundation for future human-in-the-loop work.
>
> **Scalability:** We chose Tag and RockSample because they allow reliable policy computation via SARSOP, which lets us isolate the effect of modeling suggester reliability. Extending this framework to online POMDP solvers is a natural next step and would support larger domains. Higher-dimensional latent human models or multiple latent variables are also feasible future directions. Empirically, spanning suggester behavior from low to high λ was more impactful than increasing the number of discrete types.
>
> **Reliance on Q-values:** The noisy rational suggester model uses Q-values to parameterize suggestion likelihood. Inaccurate Q-values can shift the implied suggester model, although inference remains functional and the agent still adapts to low-value suggestions. Our heuristic-suggester results demonstrate robustness to models that diverge from the rational form. In RL or online settings, an approximate or learned Q-function could be used to update suggester likelihoods. Extending this to RL in partially observable settings is feasible but belongs to ongoing research in that domain.
>
> **On the cited related work:** We thank the reviewer for pointing out “Does your AI agent get you?”. That work focuses on approximating human models from dialogue traces. While both settings involve Bayesian inference about human behavior, our goal differs. We integrate reliability as a latent variable within a sequential decision-making model and couple it with proactive querying. We will add this reference in related work.
>
> **Responses to the specific questions**
>
> *Q1.* Performance is not sensitive to the exact λ values. What matters is spanning the relevant range from low-quality to high-quality suggesters. As long as the grid covers meaningful behavioral modes, the agent adapts well.
>
> *Q2.* Inference in our model is already continuous in the sense that we maintain a probability distribution over types and update it via Bayes’ rule. Particle filtering or continuous parameterization would be another representation of the same belief update. Our choice of a small discrete latent space offers computational simplicity and worked well in practice, but more expressive continuous methods could be explored in future work.
>
> *Q3.* If the true λ lies between grid points, belief mass spreads across neighboring types, as seen in the expectation in Figure 1b. This still supports effective adaptation because in the POMDP model, the agent reasons over reliability uncertainty rather than needing to identify λ exactly.
>
> *Q4.* The ask mechanism relies on relative differences in Q-values to model suggester behavior. With approximate Q-values, the model still produces reasonable likelihoods and belief updates, though inaccuracies may shift relative weighting. When suggestions come from humans or heuristic policies, the Q-values serve as a structured prior for reasoning about their tendencies.
>
> *Q5.* Large systematic action-value errors could bias likelihood estimates. Q-values model how likely different suggestions are under each type. If that model diverges from reality, reliability estimates may shift, though with accurate Q-values the agent can still reason about reliability by grouping observable suggester behavior.
>
> *Q6.* In our discrete domains, computational cost scales linearly with the number of suggester types, since belief updates apply a Bayesian filter over a joint state with a single latent variable.
>
> *Q7.* The framework only requires a POMDP or MOMDP policy. Larger domains can be handled by online solvers or approximation methods without modifying the model. Multi-human or multi-suggester settings can be handled by introducing multiple latent variables or a structured latent model.
>
> *Q8.* Real humans introduce additional variability, but the latent reliability model is designed to capture this uncertainty. Human confidence could be integrated as an informative observation that shifts the belief over suggester types through Bayes’ rule. More refined suggester dynamics would likely be appropriate for human studies, and we plan to explore this direction.

---

### Official Review · Reviewer_b1yP · 2025-10-31

**Soundness:** 3
**Presentation:** 3
**Contribution:** 2
**Rating:** 2
**Confidence:** 5

**Summary:**

The paper studies the human-AI interaction problem under a POMDP (or MOMDP) framework. Human, as the suggester, provides occasional suggestions to the AI (autonomous agent) in a sequential decision-making environment, and the AI can utilize the suggestions to refine its belief of the underlying state. The suggestion is also captured by a quality parameter to reflect different levels of confidence during the suggestion. The levels will also be used in belief updating and thus the decision-making of the AI.

**Strengths:**

The paper is well-written and easy to follow. The model is well explained and provided with nice intuitions.

**Weaknesses:**

My main concern is the contribution of the paper:

The paper should be viewed more as a "conceptual" work. As noted above, the model is newly proposed and well-explained, but I find it hard to apply it in a real-world scenario. For the following reasons:
- Solving such a model requires knowing a lot of parameters like the transition matrix, the noisy rational suggester model, etc.
- Generally, the POMDP framework makes the model inapplicable to a real-world scenario with a moderate state space size.

The key model component is from (Asmar & Kochenderfer, 2022) and there is no algorithm specifically designed for the model.
- Would there be algorithms that can utilize the model structure to solve the problem more efficiently?
- Any theoretical guarantee for the case if the model is misspecified or the parameter wrongly estimated?

**Questions:**

See above,

---

> ### Author Response · Authors · 2025-11-22
>
> Thank you for the thoughtful comments. We address the concerns about contribution, applicability, and modeling assumptions below.
>
> **Contribution and conceptual scope of the work**
>
> Our contribution is conceptual by design, and this is the correct way to view the work. Many advances in sequential decision making begin as conceptual modeling frameworks that enable new capabilities once solvers or computational tools improve. Our framework provides such a modeling advance. It shows how to integrate latent suggester reliability and a strategic ask action into a principled POMDP or MOMDP formulation. This supports joint inference over the environment state and suggester quality and enables adaptive trust calibration and proactive information gathering, which prior suggestion-as-observation models do not support. Conceptual contributions of this kind are common and important in POMDP and planning research, where establishing a formal model often precedes specialized algorithms or real-world deployment.
>
> **On requiring transition models and suggestion models**
>
> The need for transition and observation models is shared by many planning approaches based on POMDPs or MOMDPs. Our framework does not introduce stronger assumptions than those used in prior POMDP-based shared autonomy or trust-aware planning literature. Importantly, we do not require knowing the exact suggester quality. We maintain a belief over a discrete set of types and infer reliability from interaction. This reduces the burden of specifying a fixed reliability parameter, since the agent reasons directly over uncertainty in this dimension.
>
> **On POMDP scalability**
>
> We agree that large POMDPs remain challenging, but this is a well-known and active research area. Many recent solvers such as Hyp-DESPOT, VOPP, AdaOPS, and AlphaZero-like approaches such as BetaZero expand POMDP applicability to increasingly large domains. Because our contribution is a modeling framework rather than a new solver, all of these advances apply directly.
>
> Our use of an MOMDP factorization further improves scalability relative to a naive POMDP augmentation. Observable portions of the environment state remain outside the belief, and uncertainty is restricted to a small latent variable representing the suggester type. This structure enables tractable joint inference in our experiments.
>
> **On “no algorithm designed for this model”**
> Our goal is to introduce a modeling framework that can be solved using existing POMDP or MOMDP methods. A new algorithm is not needed to demonstrate feasibility or value. We show empirically that standard solvers such as SARSOP already leverage the structure to perform joint inference over the environment state and suggester type, yielding strong performance when reliability is unknown or shifting. In this respect, our results relax the modeling assumptions used in prior work, which required known and fixed reliability, and broaden applicability to more realistic settings.
>
> **On whether specialized algorithms could exploit this structure**
>
> The value of our formulation is that it frames the problem as a standard POMDP or MOMDP. This means any advances in POMDP or MOMDP solvers apply directly without modification. By factoring the problem into observable environment state and a small latent component including the suggester type, our model aligns with existing solver architectures and benefits immediately from improvements in point-based planning, online search, or mixed-observability methods. This ensures scalability will improve as solvers advance.
>
> **On guarantees under model misspecification**
>
> Most POMDP solvers assume correct models unless the setting is explicitly formulated as robust, Bayesian, or model-uncertain. Misspecification can be handled by expanding the latent type space, by using robust or Bayesian POMDP methods, or by adopting reinforcement-learning approaches that do not require full model knowledge.
>
> Our results show that even when the suggester does not follow the assumed noisy rational structure, such as in the heuristic suggester experiment, the agent can still infer and leverage suggestion quality effectively. This demonstrates that some modeling choices are empirically robust and provides a foundation for future extensions in model-uncertain or RL-based settings.

---

### Author Response · Authors · 2025-12-02
**Author Summary for Area Chair**

We appreciate the reviewers’ time and take this opportunity to briefly summarize the paper and our responses for the area chair.

**Scope and contribution.**

This work is primarily a modeling contribution. We show how to integrate latent suggester reliability and an explicit ask action into a principled POMDP or MOMDP formulation, so that an agent can jointly infer the environment state and suggester quality and decide when to query. The aim is to provide a formal framework for adaptive trust calibration in sequential decision making, not a new POMDP solver.

**How the components fit together.**

The three main pieces serve different roles within this single framework:
1. Latent suggester types enable Bayesian adaptation to unknown and drifting reliability.
2. Dynamic type transitions model time varying suggester quality.
3. The ask action leverages this inference to query selectively when advice is likely to be valuable.

The Tables 1–5 are structured to isolate these effects and then show how they interact.

**Discretization and scalability.**

Several reviewers raised concerns about discretizing reliability and about scalability. We clarified that discretizing λ is what makes exact MOMDP inference and the use of off-the-shelf solvers like SARSOP possible, and empirically the span of λ values (poor to strong suggesters) mattered far more than fine granularity. On scalability, our contribution adds a small latent variable on top of standard POMDP structure. As with most POMDP work, the size of the base problem and the capabilities of the solver dominate. Because our formulation reduces to a standard POMDP or MOMDP, it benefits directly from ongoing solver advances and can be combined with online or approximate methods in larger domains.

**Experiments and robustness.**

The Tag and RockSample experiments are chosen because they allow reliable policies and controlled evaluation of the modeling choices. They show that the agent (i) outperforms no-suggestion and fixed-reliability baselines, (ii) adapts when suggester quality drifts, (iii) uses the ask action in intuitive high value regions, and (iv) still benefits from suggestions when the true suggester does not match the noisy rational model.

**Limitations and future work.**

We agree with the reviewers that there are important next steps, including human subjects experiments, richer latent human models, and applications in larger or continuous spaces with learned Q functions or language based suggestions. We view this paper as a step that formalizes a Bayesian trust calibration framework that can be built on by future work in human–AI collaboration and RL under partial observability.

---

### Meta-Review · Area_Chair_NaZ5 · 2026-01-04

**Summary:**

I recommend rejecting this submission. Below is a summary of the reviewers' concerns that informed this decision:
- Limited Applicability and Practicality: Reviewers found the work to be primarily "conceptual" rather than practical. The proposed framework relies on strong assumptions, such as knowing the full transition matrix and having access to a specific noisy rational suggester model. These requirements make it difficult to apply the method to real-world scenarios where such parameters are often unknown or hard to estimate.
- Scalability Issues: The experimental validation is limited to small, standard toy domains (e.g., Tag, RockSample). Reviewers expressed significant concern regarding the method's scalability to larger state spaces, higher-dimensional latent models, or continuous action spaces, which are essential for many practical human-agent collaboration tasks.
- Lack of Human Validation: Despite the paper's motivation centered on human-AI interaction and trust, there are no human-in-the-loop experiments. The absence of empirical validation with real human suggesters weakens the claims about the framework's utility in actual collaborative settings.

**Reviewer Concerns:**

### Addressed Concerns
- Clarification of Baselines: The authors clarified that no direct prior baselines exist for adaptive trust in this specific POMDP setting. They justified their comparison against fixed-trust, no-trust, and oracle agents as the relevant standard, which mitigates some concerns about the lack of extensive external comparisons.
- Robustness to Model Mismatch: Concerns regarding the strong assumption of a "noisy rational" suggester and accurate Q-values were partially addressed by highlighting the heuristic suggester experiments. The authors effectively argued (and pointed to results) showing the agent can still adapt and improve performance even when the suggester deviates from the assumed mathematical model.
- Justification for Discretization: The authors provided a reasonable technical justification for using discrete suggester types (to enable exact Bayesian filtering and compatibility with standard solvers like SARSOP) rather than continuous modeling, arguing that covering the range of behaviors is more important than fine-grained precision.

### Unsolved Concerns
- Scalability to Realistic Domains: This remains a primary objection. The experimental validation is restricted to very small, standard toy domains (Tag, RockSample) due to the reliance on exact off-the-shelf solvers (SARSOP). The authors' defense ("online solvers can handle larger domains" and "this is just a modeling framework") is theoretical. They did not provide empirical evidence that their specific joint-inference framework scales to the larger state spaces or continuous actions required for real-world human-agent collaboration.
- Limited Algorithmic Novelty: The rebuttal confirmed that the work applies existing standard techniques (MOMDP factorization, Bayesian updates) to a new variable (suggester reliability). Reviewers viewed this as an incremental "application" of known methods rather than a fundamental algorithmic contribution, a sentiment that remains unchanged.
- Lack of Human Validation: The paper is heavily motivated by human-AI interaction and "trust," yet it contains no human-subject experiments. The rebuttal explicitly defers this to "future work." Reviewers found this gap significant because real human trust dynamics and suggestion patterns likely differ from the simulated "noisy rational" or "heuristic" models used.

**Reviewer Scores:**

The reviewers are likely to maintain their original negative scores due to unresolved concerns regarding scalability to realistic domains, limited novelty, and the lack of human validation. Consequently, the consensus recommendation is to reject.

---

### Decision · Program_Chairs · 2026-01-26

Reject